# Vegetation and Environmental Changes on Contaminated Soil Formed on Waste from an Historic Zn-Pb Ore-Washing Plant

**DOI:** 10.3390/biology10121242

**Published:** 2021-11-27

**Authors:** Oimahmad Rahmonov, Jerzy Cabała, Robert Krzysztofik

**Affiliations:** 1Institute of Earth Sciences, Faculty of Natural Sciences, University of Silesia, Będzińksa 60, 41-200 Sosnowiec, Poland; oimahmad.rahmonov@us.edu.pl; 2Institute of Social and Economic Geography and Spatial Management, Faculty of Natural Sciences, University of Silesia, Będzińska 60, 41-200 Sosnowiec, Poland; robert.krzysztofik@us.edu.pl

**Keywords:** vegetation succession, rhizosphere geochemistry, mining activity, historical pollution, landscape degradation

## Abstract

**Simple Summary:**

Remnants of former Zn-Pb mining in southern Poland are an important element of geographical space. Some of the post-mining areas have found new economic and residential functions. Many of them are undergoing ecological succession and constitute valuable natural habitats enriching biodiversity of the surrounding landscapes. There are places where we can observe and document various ecological and geochemical transformations in its historical and contemporary aspects. These changes provide a basis for observing and functioning of ecosystems developing in an area transformed under the influence of Zn-Pb mining.

**Abstract:**

Post-mining waste from Zn-Pb ore exploitation undergoes processes of spontaneous succession and changes in soil chemical composition. The Zakawie area was industrially transformed by historical mining activity, ore enrichment, and the metallurgical processing of Zn-Pb ore. The subject of the study was to analyse the rate of vegetation succession (from 1999 to 2019), soil chemistry, and the relationships between them in an anthropogenic habitat with high concentrations of potentially toxic metals. Ecological and geochemical studies were carried out in an area contaminated with waste from a disused Zn-Pb ore-washing plant. Between 1999 and 2019, the transformation of grassland and meadow vegetation into scrub and forest–grassland communities was observed. This transformation led to a decrease in the area of *Molinietum caeruleae* meadow (from 25.8% in 1999 to 10.7% in 2019), whose place was taken by *Prunus spinosa* and *Rhamnus cathartica*. The community of xerothermic limestone grasslands completely disappeared, being replaced in favour of the *Diantho-Armerietum* and *Prunus spinosa* community. In this period, the share of lifeforms of plants and species composition (46 and 60, respectively) also changed. The Shannon and Simpson biodiversity index reached high values in the second investigation period, and it was 0.893 and 0.86, respectively. The anthrosols had a high content of Zn—85,360 mg kg^−1^, Pb—28,300 mg kg^−1^, Cd—340 mg kg^−1^, and As—1200 mg kg^−1^. Carbonates, clay minerals, and fe-oxides are predominant in the mineral composition of the rhizosphere; the metal-bearing phases are stable; and hardly soluble minerals include smithsonite, cerussite, monheimite, hemimorphite, and oxides of Fe and Fe-Mn. Mineralisation/crust processes formed on the epidermis, and their influences on root development were found. Scanning electron microscopy and energy-dispersive X-ray spectroscopy studies on rhizosphere soil components provide information on the type of minerals and their susceptibility to heavy metals release. The identification of some biotic and mineral structures in rhizospheres can be an interesting source of information on pedogenic processes identified in back-scattered electron images.

## 1. Introduction

From the 12th century, silver-rich lead ore was mined in the Silesia-Kraków region in southern Poland (Cabała et al., 2020). In the 19th century, the growing demand for Zn resulted in significant development of Zn-Pb ore mining. Mining and metallurgy of Zn-Pb ores in the 19th and 20th centuries has had a major impact on the natural environment in Europe. On sites where zinc, lead, copper, tin, and other metal ores are mined and processed, the most significant environmental problem is potentially toxic metals contamination of soil and water and their transfer to biotic environments [1,2,3]. Mining and the associated post-mining landscape constitute an ecological niche [4,5], where a particular succession of vegetation develops on Zn-, Pb-, Cd-, As-, and Fe-rich soils [6]. Botanical and pedological studies in areas affected by Zn-Pb-Fe ore mining have been conducted in Belgium [7], Germany [8], Spain [9], Slovenia [10,11], China [12,13,14], Morocco [3,15], and southern Poland [6].

There is also research on revegetation and reclamation of metalliferous mine wastes and other anthropogenically disturbed areas [16,17,18,19,20], where the authors showed the role of natural vegetation succession and the most appropriate methods of reclamation, revitalization, and development of post-industrial areas. The mobility and accumulation of heavy metals in soils in areas contaminated by Zn-Pb mining wastes have important natural consequences for rhizosphere chemistry and plant vegetation [21,22,23] and, as a consequence, may decide the pace of ecological processes. The chemistry of soils and their mineral composition determine the development of some plants, e.g., metalliferous ecotypes and xerothermic grasslands (e.g., calamine grasslands) [8,24,25]. These studies have shown that the processes of vegetation and soil development on individual heaps are of different natures and result from reclamation method and, age of waste disposal, and they are also related to methods of post-mining waste storage.

The period of geochemical transformations in the study area was estimated at 90 to 100 years [26]. However, in various geological situations, waste from former mining may be active, e.g., 200 years as in Belgium and the Netherlands [27], and even several thousand years in the case of waste from Roman times in England [28].

In the 19th and early 20th centuries, in an area covering approximately 1200 km^2^ between Olkusz, Bytom, and Chrzanów, hundreds of small open pits and underground mines extracting Zn-Pb ores were developed, around which metal-rich (Zn, Pb, Cd, Tl, As, and Sb) tailings were deposited. These wastes take up relatively small areas, ranging from one hectare to more than a dozen [26,29]. In these areas, specific ecosystems have been created, where spontaneous plant successions depend on the type and chemistry of the waste forming the soil’s parent rock, water balance, and climatic conditions. Relatively little data exists on the succession and transformation rates of plant communities inhabiting historic landfills [6,17]. There is also a lack of information on the primary and secondary mineral components of rhizospheres, whose chemical transformations influence the balance of toxic metals (Pb, Zn, Cd, Tl, and As) and the micronutrients (Mg, Ca, K, Na, and P) that plants need [21].

Identifying the directions of vegetation transformation in historic landfills may be helpful in designing phytoremediation in post-mining and post-smelting sites. Based on the monitoring of plant communities at 20-year intervals, and on mineralogical and geochemical studies of rhizospheres and climatic and hydrological data, an attempt was made to analyse changes in plant ecosystems in the area of a historic Zn-Pb ore-slurry disposal site. What is of interest is how quickly the plant succession changes in small ecosystems distinguished by specific pedological conditions occurred and whether the domination of new plant communities is possible. The pace of development of these ecosystems depends on the degree of geochemical transformation of minerals and potentially toxic metals in the environment [2,5,6,11]. The washing plant’s waste landfills are natural laboratories where environmental changes can be analysed in terms of geochemistry (mineral transformations and heavy metal contamination) and landscape changes as consequences of vegetation succession, especially since their formation period is well known.

The areas with post-mining waste related to the exploitation of Zn-Pb ores, as an element of the anthropogenic landscape, are subject to spontaneous vegetation succession and changes in soil properties. It is unique that the research concerned the succession of extreme vegetation contaminated with Zn, Pb, and Cd wastes habitat on an object with a 100-year period of geochemical change. The study of the relationship between vegetation and environmental conditions on anthropogenic systems is not frequent, and this article may contribute in part to addressing this gap. The aim of this study, therefore, was to analyse vegetation succession and soil contamination on an anthropogenic substrate with extremely high concentrations of potentially toxic metals, such as Zn, Pb, Cd, Mn, and Fe.

## 2. Materials and Methods

### 2.1. Study Site

The study site lies in the Silesia-Kraków region of southern Poland in an area with Mississippi-Valley-type (MVT) Zn-Pb deposits. In the open-pit mines, oxidized Zn-Pb ores (supergene type) lying close to the surface were mined in the first and sulphide ores in the deeper zones of the deposits [29].

The testing ground was the area of the Zakawie Zn-Pb ore washing plant that had been in operation in the 19th and early 20th century (50°18′36″ N 19°20′22″ E) and which is now situated within the Strzemieszyce district of Dąbrowa Górnicza (Figure 1). The waste from this historic Zn-Pb ore washing plant was deposited in the watercourse draining waters from the Zn-Pb ore washing plant. The ore washing site was situated close to an open pit Zn-Pb mine (Figure 1).

Metallurgical centres producing lead, lead oxide, and silver were already functioning several kilometres from the study site in the 12th century [29]. Zn-Pb and Pb-Ag ores occur in epigenetic Middle-Triassic ore-bearing dolomites (Figure 1). The ore-bearing horizons lie at a shallow depth and also have outcrops at the surface [29]. Limestones and dolomites of the Middle Triassic create the morphology of the area. They are locally covered by a thin layer (1–2 m) of residual weathered formations and fluvioglacial sands of the Pleistocene epoch. On carbonate formations, shallow profiles of rendzina-type soils are developed, while, on sands, there are podzolic soils. More than 100 years after the end of exploitation, anthrosols, which are locally transformed by reclamation works and artificial tree planting, were formed in the areas of ore mining and those where post-mining and wastes were stored.

The spring zone (“Źródliska” site), in which the Zakawie ore washing plant is found, has an ecological area status and an area of 1.7 ha (17,000 m^2^). Protection is the preservation of the vegetation complex in the spring zone, with valuable ecosystems (i.e., *Molinietum caeruleae*) and the localities of five plant species legally protected in Poland. Molinia meadows are protected under EU law as a natural habitat (Eu-Molinio, code: 6410) within Natura 2000 [30].

In the last 10 years, in the vicinity of the research area, several investments have been made, including digging deep ditches, which contributed to a change in water conditions. The watercourse passing through the research area has been revitalized. This procedure consisted of extending and deepening the ditch and cleaning its bottom. It also caused the lowering of the groundwater level in the protected area, and, consequently, the disappearance of the water surface and the initiation of terrestrialization processes in this area.

The actual vegetation is related to anthropogenic deformation due to urbanization, agricultural use, and especially medieval ore mining, which left a significant mark on the local ecosystems. The plant communities that dominate around the study site are of an anthropogenic nature and develop in ruderal habitats. Anthropogenic vegetation communities are mainly presented by the *Dauco-Melilotenion* union and the *Artemisietea vulgaris* class, which include nitrophilic communities of vigorous perennials and creepers in ruderal habitats. Unused arable land is subject to spontaneous succession often initiated by *Solidago canadensis*. This is an invasive species that is native to North America and that is characteristic of the *Rudbeckio-Solidaginetium* association. *Calamagrostietum epigeji* communities naturally colonize agricultural wastelands and other reclaimed areas associated with post-limestone quarry. A similar type of vegetation around the post-mining dumping site has been reported from other sites [31,32,33,34], and it was entirely different from the potential vegetation. Small-surface calcareous grasslands are developed on elevated ground and are represented by communities of the class *Festuco-Brometea*.

Part of the area around the study site was recultivated in a forestry direction by the introduction of *Pinus sylvestri, Betula pendula*, and *Robinia pseudacacia*. Attempts in recent years to afforest the area around the springs, due to difficult habitat conditions, have not been successful.

Before the drastic environmental degradation, potential natural vegetation, compatible with geological and soil conditions, developed in the studied area in the form of deciduous forests of the varieties *Tilio-Carpinetum*, *Dentario-ennephyllidis*, *Fraxino-Alnetum*, and *Leucobryo-Pinetum* on sandy habitats [35]. Today, nothing remains of these potential communities in the neighbourhood.

### 2.2. Vegetation Investigations

Vegetation studies, including plant communities and floristic investigations, were carried out in the “Zakawie Spring” ecological area in 1999 and in 2019 (Figure 2), and its area is 1.7 ha. The first stage (1999) involved detailed mapping of the vegetation in the field using various surveying methods (tape measures, theodolite, and a rangefinder) aimed at further monitoring. The investigation on the vegetation was carried out in the beginning of the growing season (May/June) and the second during the full vegetation period (July/August). The scope of the research included floristic research, vegetation mapping, and phytosociological documentation

The floristic examinations consisted of an inventory of plant species growing on the analysed area in both periods. The Shannon and Simpson biodiversity index [36] was calculated for the analysed flora. Raunkiaer’s plant life forms were defined (Appendix A). The systematic system of species was given according to Rutkowski [37]. The species names were adopted according to the study by Mirek et al. [38].

### 2.3. Vegetation Mapping

The second stage (2019) consisted of distinguishing patches of vegetation based on their colour and the range of shrub-woody and herbaceous plants on a pattern from an open-access orthophotomap, which is located in open-access spatial databases (including Google Earth [39] and the Open-Access Regional Spatial Information System (ORSIP) [40]). The obtained cartographic data was processed in the MapInfo program, which allowed for an analysis of land cover and topography.

Field surveys were carried out to verify vegetation types and actual levels of cover. The distinguished plant communities were marked on the previously prepared pattern. To identify plant communities, phytosociological documentation was consulted on individually selected areas using the Braun–Blanquet method [41]. According to this method, to identify vegetation at the level of orders, or associations, it is enough to identify the species characteristics that are distinctive for a given syntaxon.

### 2.4. Soil Investigations

Samples labelled ZKW1, ZKW 6 and ZK1, ZK4 are from a surface formed on waste from the historic Zn-Pb ore washing plant. Five kg samples were taken from the topsoil of 0–0.2 m. Samples were taken manually from an area of 0.5 m^2^. The collected material was mixed and averaged. Soil components (samples ZK1, ZK4) were separated into grain-size fractions >0.71, >0.355, >0.180, >0.09, >0.045, and <0.045 using dry methods of separation. The sampling locations are marked on the map (Figure 1). Soil samples were taken in September 2019.

### 2.5. SEM/EDS, ICP ES/MS, AAS, and XRD Analyses

Scanning Electron Microscopy (SEM) and Energy Dispersive X-Ray Spectroscopy (EDS). Microanalyses were conducted to identify stable and unstable mineral phases that may be sources of metal ions. Soil components from rhizospheres and root fragments were collected by hand and glued onto carbon tape. Over 30 specimens were tested, and 120 back-scattered electron (BSE) images and over 200 EDS analyses were obtained. These were carried out using a Philips XL 30 scanning electron microscope with an EDAX analyser. BSE images were obtained using a Centaurus attachment, with a detector resolution of 0.3 Z. The accelerating voltage was 15 kV. These analyses were performed in atmospheric- and low-pressure (0.3 Torr) modes. The EDS spectra analyses were processed using Phillips software.

ICP-ES/MS. The concentrations of selected elements were determined in the soils (ZKW 1, … ZKW … 6) collected from the rhizosphere zones. The MA270 procedure (Bureau Veritas Minerals Laboratories in Canada) was used based on the MULTI-ACID ICP-ES/MS method.

AAS. Heavy metal contents (Zn, Pb, Fe, Mn, Cd, and Tl) were analysed by atomic absorption spectroscopy (AAS) using a SOLAAR M6 spectrometer. The sample soils were averaged and dried. Samples (0.2 g) were ground in an agate mortar. A mixture of pure acids was used for the mineralization of each sample: 40% HF (2 mL), 65% HNO3 (3 mL), and 35% HCl (1 mL) and distilled water (2 mL). The mineralization was carried out at (110 °C) in a Milestone MLS 1200 microwave furnace. To remove fluorosilicates, 50 mL of 4% H3BO3 was added and each sample again mineralized for a short time. The resulting solution was transferred to a 100 mL flask and filtrated under pressure to plastic bottles using 0.45 μm filters.

X-ray Structural Studies. The mineral composition of the soils was identified using X-ray powder-diffraction (XRD) methods. XRD data were obtained using a PANalytical X’PERT PRO–PW 3040/60 diffractometer (CuKα1 source radiation, Ni-filter to reduce the Kβ radiation, and an X’Celerator detector). Quantitative data processing was performed by means of the X’PERT High Score Plus software using the latest PDF4+ database. Chemical (AAS) and mineralogical (XRD) studies were conducted in the laboratories of the Faculty of Natural Sciences at the University of Silesia.

## 3. Results

The processes of vegetation range transformation here are related to water relations (Table 1, Figure 2). The change in humidity caused the withdrawal of hydrophilic species and protected species as (Appendix A), which were the basis for the establishment of the protected area.

### 3.1. Initial Stage of Vegetation Succession

In 1999, the secondary succession was distinguished in the study area in two stages: initial and optimum. The initial stage was located in the western and central part of the site and was marked on the map as a vegetation-free area (Figure 2, 1999). These were sandy surfaces and rocky debris (from the washing plant Zn-Pb ore). Such surfaces were adequately colonised by *Populus tremula*, *B. pendula*, and accompanying pioneer *Koelerio-Corynephoretea*-class psammophilous species. The areas occupied by these species in 2019 were devoid of vegetation in 1999 (Figure 2, 1999; Table 1, 1999), and were in the initial stage of succession.

The *Pinus sylvestris* vegetation only encroached after 1999 and now forms a dense pine thicket with 15-year-old specimens. The bottom of the watercourse is currently overgrown by *Caricetum paniculatae* (Figure 3), accompanied by single specimens of *Cirsium rivulare*, *Carex remota*, and *Caltha palustris*. In this section, the watercourse has a straight character, and its banks are eroding and widening the bottom of the watercourse, which created additional ecological niches for hygrophilous species.

### 3.2. Vegetation Changes

There have been fundamental changes in the range of the vegetation community during the analysed period (1999–2019). Surfaces associated with sand grasslands were replaced mainly by shrubs of *Prunus spinosa* and *Rhamnus cathartica*, *Armeria maritima*, and species from the *Arrhenatherion elatioris* union (Figure 2). The process of transformation of one phytocenosis into another in a simplified form is shown in Figure 4.

Over a period of 20 years, fundamental and dynamic changes took place in the vegetation (Figure 2). These mainly concern the *Molinietum caeruleae* community. It occupied 25.8% (4386 m^2^) of the site area in 1999, while in 2019 it was only 10.7% (1819 m^2^, Table 1). These areas (after *Molinia*) have been colonised by the community with Armeria maritima and the shrub *Rhamnus cathartica*. The succession here goes towards the formation of thermophilic communities of the class *Rhamno-Prunetea*.

It is interesting to note the transformation of xerothermic limestone grasslands, which have completely disappeared in favour of the *Diantho-Armerietum* community and *Libanotis pyrenaica* (southeast of the site, Figure 2).

The xerotermic limestone grasslant with 28.2% surfece (in 1999) changed mainly into to *Libanotis pyrenaica* community (Figure 2 and Table 1)—in the eastern part of the study area (Figure 2).

Significant changes were found in the range of the *Arrhenatherion elatioris* community, where its area decreased from 21% to 5.1% (Table 1). Such changes were caused by changes in moisture relationships and the encroachment of tree and shrub species, which led to changes in light conditions, such as insolation. It was replaced by a community composed of *R. cathartica*, *Prunus spinosa*, and *A. maritima* (Figure 2).

As a result of a 20-year succession of vegetation, a dynamic development was observed in the formation of thermophilous periwinkle communities with *Prunus spinosa* and *Rhamnus cathartica*. In 1999, their encroachment was observed (specimens were about one metre high and occurred singly), while, in 2019, some specimens were about two metres high. *P. spinosa* and *R. cathartica* shrubs form dense thickets, and, under their canopies, *Cornus sanguinea* and *Rosa canina* occur singly. Taking into account that they thrive on shallow calcareous soils, the terminal stage of succession will be assemblages of the *Rhamno-Prunetea* class.

Wellspring areas—significant changes take place here in both water relationships and aquatic biocoenosis. The water surface of 272 m^2^ (1.6% of the study area) from 1999 disappeared and with it the biocoenoses described at that time: *Batrachium aquatile* and *Sparganium ramosum* (Figure 3 and Figure 5). Currently (2019), this phytocenosis has been changed into to sedge community with *Carex gracilis*, *C. riparia*, *C. rostrata I*, and *C. canescens*. Other changes in the area of communities and other landscape elements are presented in Table 1 and Figure 2, 1999 and 2019.

### 3.3. Flora Analysis

The flora changed notably in the analysed period. In 1999, the occurrence of 46 vascular species was recorded, while, in 2019, 60 species were recorded. The found species belong to 33 families and 62 genera (Appendix A). The richest in species were the *Compositae*, *Rosaceae* and *Cyperaceae* families (Table 2). The Shannon and Sipmson biodiversity index reached high values in the second investigation period, and it was 0.893 and 0.86, respectively. Changes were also observed in the plant life forms, which mainly concerned a decrease in the number of hydrophytes (from 14% in 1999 to 7.8%—2019) and geophytes (from 14.03% to 6.3%) and an increase in nanophanerophytes (Table 2)—such changes were due to the water relations.

Beyond the wellspring zone, along the watercourse, willow thickets were formed with *Salix cinerea, S. caprea, F. alnus, Euonymus europaeus, Corylus avellana, Rhamnus cathartica, Prunus domestica, Viburnum opuls*, and specimens of *Sambucus nigra, Padus avium*, or *Padus serotina*. Among the herbaceous plants occurring here were: *Geum rivale, Caltha palustris, Cirsium rivulare, C. oleraceum, Geranium pratense, Carduus nutans,* and *Valeriana officinalis*. These are fragments of the remaining *Cirsietum rivularis* community shown in Figure 2, 1999.

On slopes exposed to sunlight, on calcareous and sandy substratum, and in the open parts of the site, there are typical xerothermic species such as *A. maritima subsp. halleri, Dianthus deltoides, D. carthusianorum, Festuca ovina, Hieracium pilosella, Plantago media, P. lanceolata, Thymus pulegioides, T. serpyllum, Sedum acre, Rumex acetosella, Euphorbia esula, Ornithogalum umbellatum, Gallium mollugo, Agrimonia eupatoria, Scabiosa ochroleuca, Centaurea stoebe, Centaurea scabiosa,* and *Falcaria vulgaris*. A significant difference in relation to 1999 was observed with regard to rare and protected species. In 1999, *Dactylorhiza maculata, Epipactis palustris, Iris sibirica, Gentiana pneumonanthe, Gladiolus imbricatus*, and *Ophioglossum vulgatum* were found on the site. These species were not found in 2019. However, *Listera ovata* was found, which had not been previously reported.

### 3.4. Potentially Toxic Metals’ Concentration in Soil

On historic mining sites (from the 12th to 20th centuries), in areas with shallow Silesia-Kraków Zn-Pb ores, soils were often formed on former post-mining waste sites or waste from other washing processes. A characteristic presence in the geochemistry of these soils is a distinctive group of elements related in origin to Zn-Pb ores. These are heavy metals such as Fe, Zn, Pb, Mn, Cd, and Tl, as well as the metalloids As and Sb. The dolomitic-limestone rocks hosting Zn-Pb ores means that the soils are also enriched in microelements, which influence the conditions of plant development (Al, Ca, Mg, K, Na, and P). The frequency with which such elements occur in the soils of the Silesia-Kraków region is presented in the histogram (Figure 6).

The concentration of potentially toxic metals in the soil formed on waste from the historic Zn-Pb ore washing plant reached very high levels (Table 3). Particularly high contents were recorded for zinc (85,360 mg kg^−1^), lead (28,300 mg kg^−1^), and cadmium (340 mg kg^−1^). The levels of manganese (12,020 mg kg^−1^) and arsenic (1200 mg kg^−1^) were also high and elevated for thallium (9.8 mg kg^−1^). Among other metals studied, e.g., Fe, Cu, Ni, and Ag, no contents were found that differed from the average for Polish and European soils [43,44]. The soils w rich in iron, calcium, magnesium, aluminium, and potassium and relatively low in phosphorus and sodium (Table 3). The most toxic heavy metals are: Pb, Cd, Tl, and the metalloids As and Sb [43]. Zinc has limited toxicity only in very high concentrations and in interaction with cadmium [1,2,12,42].

Potentially toxic metals (Zn, Pb, Fe, Cd, and Tl) bound in minerals derived from primary Zn-Pb ores occurred in all fractions of the soils investigated. The level of potentially toxic metals content did not depend on the size of the fraction (Table 4).

The levels of Zn, Pb, and Cd concentrations were very high and exponentially exceeded the permissible standards for soils. High cadmium concentrations are hazardous; for some soils, e.g., ZK4 in the fractions studied, they ranged from 195 to 495 mg kg^−1^ (Table 4). Toxic thallium of 8.3 to 9.9 mg kg^−1^ was identified only in ZK2 (Table 4).

Analysis of mineral composition can indicate whether heavy metals can leach easily into soil solutions and groundwater (e.g., from non-permanent minerals such as sulphates and sulphides) or whether they are bound in permanent minerals (e.g., silicates, oxides, and carbonates).

### 3.5. SEM Data

The analysis of BSE images and EDS and XRD spectra indicated that the soils from the Zakawie washing plant area are composed mainly of minerals originating from the waste produced in the Zn-Pb ore enrichment process (Table 5). In the mineral composition of these soils, it is possible to distinguish groups of minerals differing in origin.

Among the minerals present in rhizospheres, the most abundant were aluminosilicate mineral aggregates: quartz, dolomite, feldspars, and Fe and Fe-Mn oxides (Table 5; Figure 7A–C). These minerals are the main components of the rhizosphere soil. The proportion of Fe and Fe-Mn oxides was particularly high. Polymineral aggregates of Fe oxides contain zinc in their structure due to the presence of submicroscopic Zn carbonates or sorption of Zn^2+^ ions on aluminosilicates and Fe oxides (Figure 7A–C). The presence of zinc was marked in many EDS spectra of the aggregates of aluminosilicates, dolomite and Fe oxides (Figure 8B–D), and Pb carbonates that were studied (Figure 7E).

There was a significant proportion of Pb carbonate and cerussite, with grains ranging in size from a few micrometres to 50 μm (Figure 7A,C–E). Grains of Zn carbonate and smithsonite were identified in the rhizospheres, while the Zn-Fe carbonate, monheimite, and Zn silicate and hemimorphite were identified much less frequently (Figure 7A,F). Other accessory metal-bearing minerals were very rare in the rhizosphere, and these were single grains of Ti-oxide, Ti-Fe oxide, Fe-Cr oxide, Ce, and La phosphate with REE (Figure 7G).

On the epidermis of the roots of plants (*M. caerulea*) inhabiting the area of the historic washing plant, crusts are often formed from submicroscopic grains of Fe oxides, Zn and Pb carbonates, quartz, and dolomite bonded with aluminosilicates, which are mainly from the group of clay minerals. In aluminosilicate coatings enveloping the roots, metal-bearing grains can easily be distinguished: Pb carbonates and Zn carbonates (Figure 8C,D).

There was a high proportion of Fe oxides, which were enriched in Zn (Figure 8C). The surface of some roots was completely covered, impermeable, and with a thickness between 5 and 20 μm. Carbonates and oxides rich in Zn, Pb, and Fe were present en masse on the epidermis. These minerals are stable under oxidative conditions, with an excess of carbonate, Zn^2+^, and Pb^2+^ ions that are not leached into soil aqueous solutions. In this situation, the leaching of metals (Zn, Cd, and Pb) from carbonate phases, oxides, or aluminosilicates is possible. Analysis of BSE images (Figure 7C,D) showed that some metal-bearing phases had traces of dissolution and leaching; therefore, it can be assumed that under specific conditions in rhizospheric zones, limited transfer of Zn, Cd, and Pb ions into solutions is possible.

## 4. Discussion

### 4.1. Vegetation Changes

Due to the topographical differences of the site, the degree of succession and the flora showed considerable diversification in terms of ecological requirements. As long as the springs had high efficiency, the area was largely covered with water (Figure 5), and there were hydrophilic and hygrophilous species (Table 1 and Appendix A). Changes in water relations in the study area were related to the anthropogenic impact of human activity. In recent years, due to long-lasting droughts, lowering the groundwater level, the springs and the stream operate periodically when there is heavy rainfall. This contributed to the disappearance of aquatic habitats and related flora (Appendix A). Changes are evident in the numbers of plant species (Table 2) from 1999 (46) to 2019 (60). These changes also confirm the Shannon and Simpson diversity index. The water surface was therefore a physical barrier, and its disappearance favoured the encroachment of other species and the acceleration of overgrown processes resulting from the fertile of bottom habitats. Similar regularities were found in coal storage sites in Europe [5,17,31,32,33,34,45]. Hence, paradoxically, it can be said that the water deficit promotes the acceleration of succession by entering mainly shrubby xerophilic species that do not have high ecological requirements.

The formation of diverse habitats has a favourable impact on the rate of succession stages, even in small areas. In the structure of non-forested communities, young specimens and seedlings of woody species such as *Alnus glutinosa P. tremula, B. pendula*, and *P. sylvestris* appear, which are evidence of the forest direction of succession, which is the terminal stage in a temperate climate. Sedge rushes and typical calcareous grasslands with *Libanotis pyrenaica* are also formed in Zakawie. Such grasslands are formed by natural succession and are abundant in species resistant to potentially toxic metals stress. In the area studied herein, such species may include *Armeria maritima* and *D. carthusianorum*. Given the lack of historical data on the original vegetation in this area, its physiognomy may be inferred by analysing habitat conditions in zones undisturbed by humans.

Reclamation work and ploughing of the western section of the site resulted in the partial destruction of the developing xerothermic limestone grasslands (Figure 2). Ploughing also had a positive impact, as the waste from the washing plant was mixed with lower-lying sands, thus creating a subsoil with better conditions for vegetation. Over a period of 10–15 years, communities with galmanic plants, including the *Diantho-Armerietum* complex, were newly formed on this subsoil. Similar vegetation transformations were recorded by other authors [6] in similar habitat conditions.

Spontaneous ecosystem restoration in post-mining areas usually occurs very slowly, especially in soils deficient in plant nutrients (P, N, Mg, K, and Na) and with low water capacity. This is conditioned by the chemistry of the waste, and high concentrations of toxic metals (Zn, Pb, and Cd) limit the development of microorganisms involved in transforming organic matter and those indirectly involved in soil formation [16,46]. In the case of post-mining waste in Zakawie, the succession was relatively fast, despite the high concentration of toxic metals in the substrate. It should be emphasized that this waste was deposited over 100 years ago, and, at that time, it became unavailable. Such processes were described in the literature [2].

To a limited degree, under conditions conducive to the decomposition of organic matter with the involvement of bacteria and fungi, pH can be lowered, and microenvironments can be formed under reducing conditions [26]. A high content of heavy metals in plants, soils, waters, and sediments is often observed near mining or ore-processing sites [15,29,47], which plays an important role in the course and rate of succession [17,35] and also in the floral composition of brownfield sites [5,6,48,49].

The Zakawie ecosystem differs from other ecosystems formed in areas of shallow exploitation of supergene ore deposits, e.g., Bolesław, Poland [6,25]; Plombiers, Belgium [7]; or the Harz Mountains of Germany [8].

The cover of metal-rich waste from Zn-Pb ore processing ranges from 20 to 80 cm. Xerothermic grasslands developed only in the central part of the area on small hills, where the thickness of sediments is the highest. The high proportion of clay minerals, iron oxides, and carbonates increases the pool of microelements: K, Na, Ca, and Mg. It also improves the soil’s ability to retain water, which limits its erosion.

The study area w dominated by communities of *Armeria maritima*, *Dianthus deltoides,* and *D. carthusianorum*. Lowering of the water-bearing horizon related to lower productivity of the “Zakawie” spring worsens the habitat conditions. This favours the development of species with diversified ecological requirements (Table 1 and Table 2). Wetland fragments are occupied by representatives of hydrophytes, hygrophytes, and mesophytes. A similar relationship of plant succession was described in the USA (Massachusetts) [50].

The history of the degradation of the original soil cover and plant communities in the area goes back several centuries. It is directly related to medieval Pb-Ag mining [29] and, from the 19th century to 20th century, Zn-Pb mining. Over the last few decades, ecosystem changes in these areas have been associated with revitalisation through artificial plantings. Wooded areas with species resistant to environmental pollution (*B. pendula*, *Larix decidua,* and *Robinia pseudacacia*) were created. An analysis of the condition of artificial plantings around the area in question proves that these attempts were unsuccessful. Much-better specimens of trees and their communities are the result of spontaneous succession. Approximately 200 m to the west of the area studied is a wooded waste heap. It is a potential biochora, providing propagules to the site, and it increases the rate at which woody species encroach [5,45].

In a study by Szarek-Łukaszewska [6] near the Olkusz region, it was shown that he development of trees on a 100-year-old site was retarded by atmospheric pollution (SO_2_, metallic dusts) emitted from smelting plants adjacent to the mine. When these emissions rapidly declined in the 1990s, pine trees began to multiply in the 100-year-old grassland. At the beginning of the 21st century, atmospheric deposition of nitrogen increased in the investigated area, as in other parts of the country [51]; this will undoubtedly lead to eutrophication of habitats and consequent changes in the vegetation and soil. Thus, over time, the influence of post-mining areas’ metal pollution on the vegetation cover decreases, and the dominant role in the direction and rate of plant successions is due to other environmental factors, which is similar to non-polluted areas.

### 4.2. Geochemical and Mineralogical Transformation of Historical Waste

In the study area (Figure 1 and Figure 2), soils were formed on metal-bearing waste from the Zn-Pb ore enrichment process. SEM/EDS and AAS studies indicate that differences in soil mineral composition and potentially toxic metals concentration levels are discernible in different fragments of the study area. On this surface, spontaneous vegetation successions occurred in topsoil with extremely high metal concentrations. Concentrations of Zn reached 85,360 mg kg^−1^; Pb was up to 28,300 mg kg^−1^; and Cd was up to 340 mg kg^−1^ (Table 3). In some fractions, the metal concentration levels were even higher (Table 3). The overall content of metals in the soil is only indirectly related to the toxic effects on plants and most often, they are in an unavailable form [2]. The presence of calcium ions also partially neutralizes the toxic effects of heavy metals. The mineralogical data obtained in this study confirm this phenomenon.

Over a period of more than 100 years, chemical changes occurring in the waste and the soils developed on it have led to complete oxidation of primary Pb, Zn, and Fe sulphides (Table 5). Therefore, no unstable Zn, Pb, and Fe sulphides were identified in the rhizosphere, which could be the source of Pb^2+^, Zn^2+^, Cd^2+^, and SO_4_^2-^ ions transferred to soil aqueous solutions. The weathering processes of limestone and dolomite regulate the soil reaction that limits the migration of Zn, Pb, and Cd, which are bound in situ in carbonates and Fe oxides that are hardly soluble. The absence of sulphides of Zn, Pb, and Fe means that SO4^2-^ ions are not released from them, which could in turn lower the pH of soils and increase the pool of potentially toxic metals.

A sulphur concentration below 0.1% max. 0.27% is associated with barite and gypsum. Zinc, lead, and cadmium are bound in stable minerals; carbonates (smithsonite and cerussite), silicates (hemimorphite), and oxides (goethite and lepidocrocite) have the possibility of leaching of potentially toxic metals, which is markedly limited [52]. Zinc has been identified in Fe-oxide aggregates, and aluminosilicates are bound in submicroscopic carbonate phases. The sorption of Zn^2+^ ions on oxides and aluminosilicates can also result in zinc concentrations [53].

Metals (Zn, Pb, Cd, Fe, and Mn) and the metalloids As and Sb in the soils studied are bound in secondary minerals such as:-Lead in cerussite.-Zinc in smithsonite, monheimite, and hemimorphite.-Iron and manganese in oxides and hydroxides of Fe and Mn.-Cadmium (not present in its own phases) is bound to Zn carbonates.-Arsenic and antimony are bound in Fe oxides/hydroxides.

In an environment with an excess of carbonate ions, such as soil formed on limestone, these minerals are stable and have limited solubility, hence the small pool of bioavailable lead, cadmium, and zinc. High total concentrations of metals (Table 3 and Table 4), bound in stable, insoluble minerals, do not hinder the development of many plant species. However, the processes of forming polymineral crusts on roots may limit the growth of some plant species, particularly those whose root zones develop exclusively in metal-contaminated soil. Plants with deeper roots, such as *Prunus spinosa* and *Rhamnus cathartica*, develop in the deeper layers of the former washing plant area, have a longer vegetation period, and displace other plant species such as *A. maritima subsp. halleri, Dianthus deltoides, D. carthusianorum, Festuca ovina, Plantago media, P. lanceolata, Thymus pulegioides, T. serpyllum,* and *Sedum acre*. This relationship is clearly visible on small hillocks formed from waste, which are overgrown only with plant species adapted to growth on metal-bearing heaps (*A. maritima subsp. halleri, Dianthus deltoides,* and *D. carthusianorum*). A similar floristic composition and plant communities have been described in neighbouring areas of similar genesis [6,25] and are called calamine grasslands.

Among the species analysed, *Molinia caerulea* reveals interesting adaptations due to its root topology, which is dependent to some degree on the quality of soil and its humidity. Changes in water conditions contributed to the reduction in *M. caerulea* populations in the area studied, which consequently has an effect on other hygrophilous species, and led to a change in the direction of succession from grassland to forest-grassland.

One of the characteristics of this species is its ability to utilise nutrient-rich soils through changes in root morphology [54]. Roots can adapt to soil heterogeneity by changing their topology. The roots of *M. caerulea*, a dominant species in the area analysed, can reach depths of more than 40 cm, and, in other regions, a depth more than 80 cm was noted [55]. Their topology also shows high plasticity. The root morphology of this species can change as a result of mineralisation of the root epidermis by polymineral aggregates of aluminosilicates, Fe-oxides, and Zn and Pb carbonates (Figure 8A–D). Roots and absorbent hairs that are completely surrounded by mineral crusts lose their function. The plant responds by producing new, additional roots (Figure 8B). This guarantees continuous growth and enables the plant to take over new areas (Figure 2/Table 3) under extreme habitat conditions, as well as those under potentially toxic metals stress or nutrient deficiency [52].

The interaction between plant roots, solutions, and minerals of the rhizosphere always occurs, and its intensity depends on plant physiology and soil chemistry. Root secretions interact with minerals and soil solutions and thus influence the rate of mineral dissolution. In soils contaminated by metal ore smelters and mines, the rhizosphere chemistry affects the migration of metals such as Al, Ca, Mg, Cd, Ni, Pb, and Zn [21,26]. Submicroscopic examination of the root epidermis or the immediate root zone makes it possible to identify secondary carbonate and silicate minerals, which have a significant influence on soil chemistry, especially in areas with extreme concentration of trace elements.

Changes in soil solution chemistry can be assessed from studies of unstable, metal-bearing mineral phases [21,56,57]. However, there is insufficient knowledge about the interaction between root secretions, microorganisms, and the formation of secondary metal-bearing phases in rhizospheres. The secondary metal-bearing mineral phases identified on the root epidermis demonstrate that metal ions such as Zn, Pb, Fe, and Mn are present in soil solutions. However, these metals are mostly bound in stable, insoluble Zn and Pb carbonates and Fe and Mn oxides. Metal-ore-mining and -processing areas are a good testing ground for studying the relationship between highly metal-contaminated soil and the ability of plant communities to vegetate. Chemical transformation of minerals and leaching of metals into soil solutions are active even a century after mining activities have ceased [58,59]. Information about areas of former mining activity is rapidly disappearing [59]. These areas were numerous and occupied large areas during the period of intense ore mining from the early 19th century to the mid-20th century [60]. Results from interdisciplinary botanical and geochemical studies in these areas can be used to identify historic metal ore-mining and -processing sites.

The variability in the thickness of the washing plant waste cover, the presence of a spring zone, and good exposure to sunlight meant that despite extreme environmental conditions, the process of vegetation succession in this area takes place in a relatively short time (20 years). This is much faster than in post-galmanic areas from the nearby Olkusz area described in other research [6,61,62,63] or in the abandoned lignite-mining area of Goitsche, Germany [64]. Over a period of more than 20 years, shrubs, e.g., *Prunus spinosa* and *Rhamnus cathartica*, have effectively replaced xerothermic grasslands in the Zakawie area. This is the consequence of spontaneous succession and improvements in habitat conditions, which favour the growth of trees, but mainly shrubs, resulting in the shading of swards and withdrawal of light- and heat-loving species (Table 1). During the growing season, a crust composed of aluminosilicates, feldspars, and iron and manganese oxides with a few zinc and lead carbonates forms on the roots. Maintaining these kinds of swards in post-mining areas is advisable due to valuable plant species and their biotic diversity. However, in many cases, active protection is required.

## 5. Conclusions

The research results presented in this article are an important contribution to the discussion on the role of Zn-Pb contamination in a heavily polluted environment. They address key triple relationships: historical metal-ore processing and mining, contemporary environmental changes, and restoration methods. The results of the study show that:Despite its small size, the area of the historic Zn-Pb ore washing plant has a varied relief and soil substrate, as well as a disappearing watercourse, which affects the heterogeneity of the habitat and the diversity of the flora in terms of ecological requirements.In comparison to 1999, there was an increase in the number of species, families, and genera in 2019. The lifeforms of plants have changed. The share of nanophanerephytes and hemicryptophytes has significantly increased, while the share of geophytes, hydrophytes, and therophytes has decreased.Changes in water relations affected the composition of the flora, especially the range of *Molinia cereluea*. Over a period of 20 years, a transformation of meadow vegetation (*Molinietum caeruleae*) into scrub and forest-plant communities took place. The development of thermophilous shrub communities with *Prunus spinosa* was recorded during this period of succession.Through SEM/EDS studies of the root zones, especially the mineral components, additional information can be obtained regarding the occurrence of potential toxic elements, as well as microelements important for vegetation. Furthermore, identifying certain biotic and mineral structures in the rhizosphere can provide interesting information on pedogenic processes.Mineralisation of plant roots, especially the formation of impermeable crusts on the epidermis can significantly impede vegetation, and plants that have the ability to sprout new trichomes can increase their ability to vegetate.The complete absence of primary sulphides of Zn, Pb, and Fe in the waste indicates that after 100 years their oxidation processes have reached an advanced stage. Despite the extreme contents of potentially toxic metals, vegetation development is not disturbed because the pool of bioavailable Zn, Pb, and Cd is limited due to their binding in immobile carbonate phases, oxides, and silicates.

The results obtained can be used in the reclamation of post-mining areas and for ecological risk assessment in land-use planning and management of neighbouring areas.

## Figures and Tables

**Figure 1 biology-10-01242-f001:**
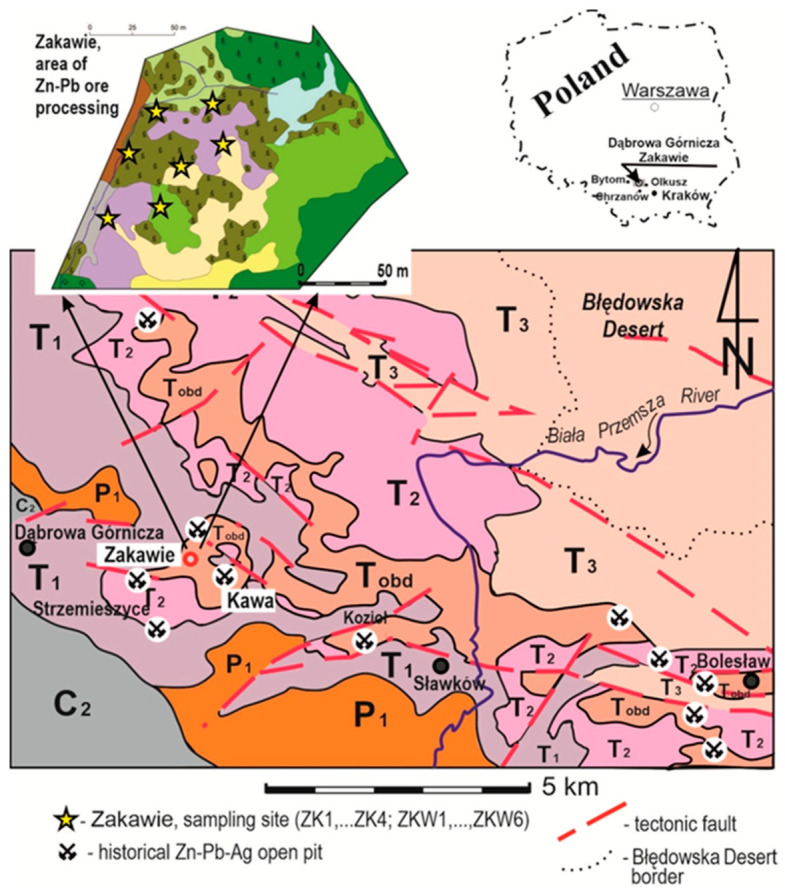
Geological sketch of the study area. Legend: C_2_—UPPER CARBONIferous—USCB boundary, P_1_—Lower Permian, T_1_—Lower Triassic, T_2_—Middle Triassic, T_3_—Upper Triassic, T_obd_—ore-bearing dolomites with Zn-Pb-Ag mineralization [29].

**Figure 2 biology-10-01242-f002:**
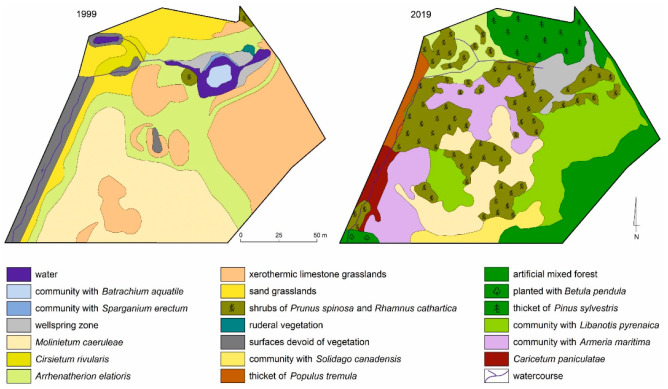
Vegetation distribution in 1999 and 2019.

**Figure 3 biology-10-01242-f003:**
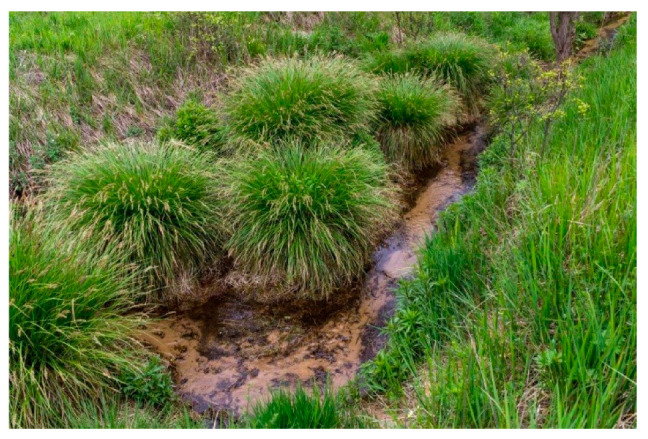
Community of *Caricetum paniculate* (2019).

**Figure 4 biology-10-01242-f004:**
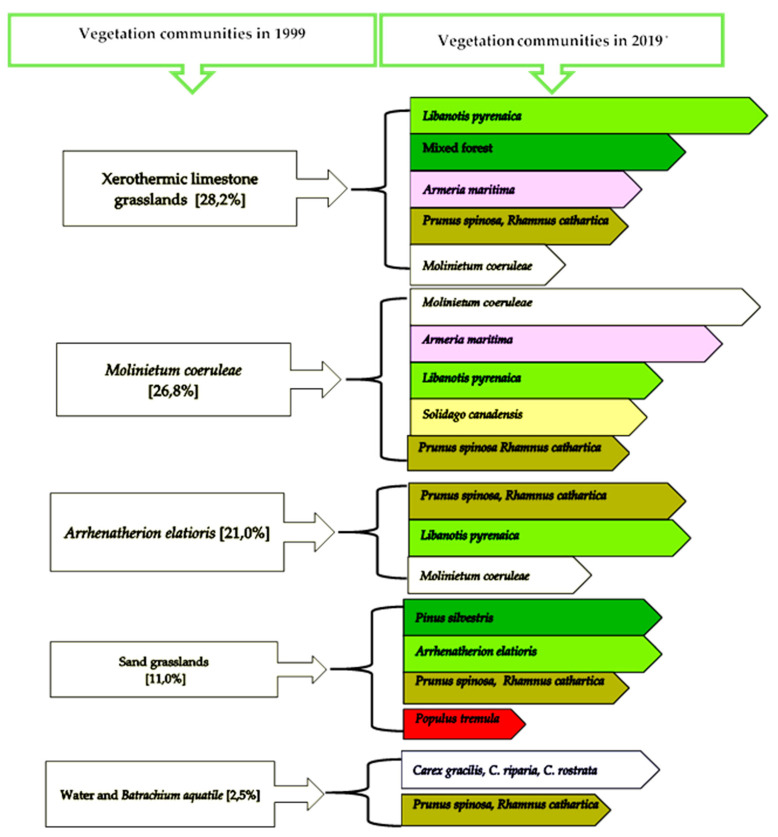
Main courses of vegetation changes from 1999 to 2019 in Zakawie region. * See Table 1 for the percentage share of each community.

**Figure 5 biology-10-01242-f005:**
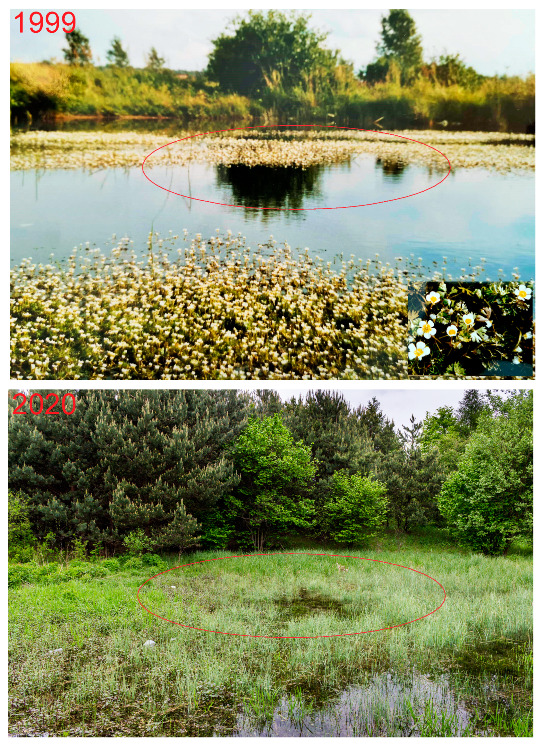
1999: water surface and community with *Batrachium aquatile*; 2020: wellspring zone with sedge rushes.

**Figure 6 biology-10-01242-f006:**
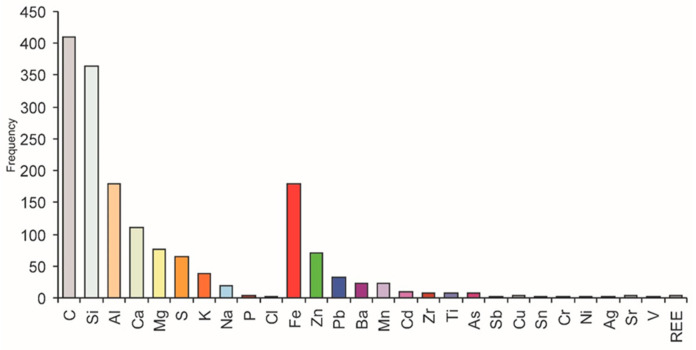
Relative enrichment of various elements in the topsoil. Areas affected by Zn-Pb ore mining in the Olkusz-Dąbrowa Górnicza region (southern Poland). Based on 800 SEM/EDS analyses after Cabała [42].

**Figure 7 biology-10-01242-f007:**
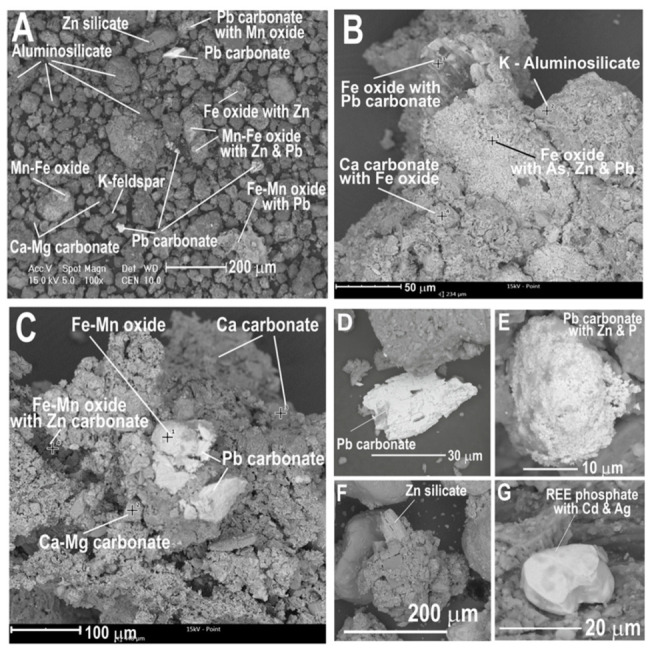
BSE images of minerals from Zakawie rhizosphere plants.

**Figure 8 biology-10-01242-f008:**
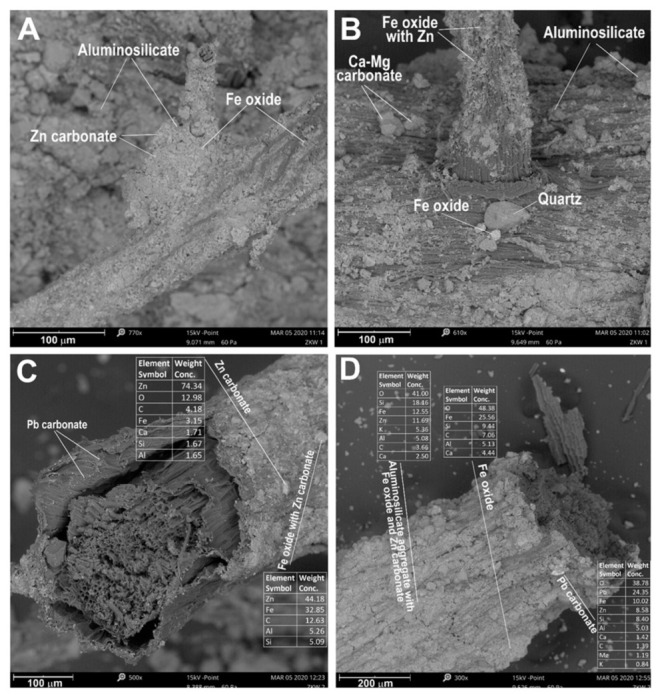
BSE images of the root epidermis *M. caerulea* from Zakawie.

**Table 1 biology-10-01242-t001:** The areas inhabited by plant species in 1999 and 2019.

No.	Vegetations	Surface in 1999 (%)	Surface in 2019 (%)
1.	*Molinietum caeruleae*	26.8	10.7
2.	*Arrhenatherion elatioris*	21.0	5.1
3.	Shrubs of *Prunus spinosa* and *Rhamnus cathartica*	0.7	22.3
4.	Wellspring zone	1.9	3.7
5.	Xerothermic limestone grasslands	28.2	-
6.	Sand grasslands	11.0	-
7.	Surfaces devoid of vegetation	6.7	-
8.	*Cirsietum rivularis*	1.7	-
9.	Water	1.6	-
10.	Community with *Batrachium aquatile*	0.9	-
11.	Ruderal vegetation	0.3	-
12.	Community with *Sparganium erectum*	0.2	-
13.	Community with *Armeria maritima*	-	12.7
14.	Community with *Libanotis pyrenaica*	-	18.5
15.	Artificial mixed forest	-	10.9
16.	Community with *Solidago canadensis*	-	3.6
17.	Thicket of *Pinus sylvestris*	-	7.6
18.	*Caricetum paniculatae*	-	2.3
19.	Thicket of *Populus tremula*	-	1.7
20.	Planting with *Betula pendula*	-	0.9

**Table 2 biology-10-01242-t002:** Selected features of analysed flora in Zakawie.

Family	1999	2019
*Compositae*	4 *	7
*Rosaceae*	2	7
*Poacea*	6	5
*Cyperaceae*	3	5
*Lamiaceae*	2	4
*Caryophyllaceae*	2	2
*Orchidaceae*	2	1
*Plantaginaceae*	3	3
*Salicaceae*	2	3
*Adoxaceae*	1	2
*Apiaceae*	2	2
*Betulaceae*	0	2
*Caprifoliaceae*	1	2
*Juncaceae*	0	2
*Ranunculaceae*	2	1
*Rhamnaceae*	2	2
Raunkiar’s life forms (%)		
Megaphanerophyte (M)	3.5	6.3
Nanophanerophyte (N)	10.5	18.8
Herbaceous chamaephyte (C)	5.3	4.7
Hemicryptophyte (H)	45.6	56.3
Geophyte (G)	14.03	6.3
Therophyte (T)	7.01	0.0
Hydrophyte (Hy)	14	7.8
Shannon biodiversity index	0.633	0.898
Simson biodiversity index	0.761	0.861

*—number of species per family.

**Table 3 biology-10-01242-t003:** Elemental composition of the topsoil from Zakawie.

Sample Number	Cu	Pb	Zn	Ag	Ni	Mn	Fe	As	Sr	Cd
	mg kg^−1^	mg kg^−1^	mg kg^−1^	mg kg^−1^	mg kg^−1^	mg kg^−1^	%	mg kg^−1^	mg kg^−1^	mg kg^−1^
ZKW 1	21	1860	>10,000	1.7	76	4430	6.1	390	53	320
ZKW 2	16	>10,000	>10,000	1.2	27	1040	3.7	380	56	140
ZKW 3	18	4130	>10,000	0.6	57	2340	5.6	380	49	290
ZKW 4	32	>10,000	>10,000	3.3	43	1770	6.3	730	59	190
ZKW 5	49	26,780	85,360	2.4	41	1530	5.5	620	53	340
ZKW 6	32	28,300	45,250	1.4	52	12,020	10.2	1200	46	180
	**Sb**	**Ca**	**P**	**Mg**	**Ba**	**Al**	**Na**	**K**	**S**	**Tl**
	**mg kg^−1^**	**%**	**%**	**%**	**mg kg^−1^**	**%**	**%**	**%**	**%**	**mg kg^−1^**
ZKW 1	1.6	6.9	0.05	3.6	170	3.4	0.05	0.9	<0.1	9.8
ZKW 2	1.4	7.0	0.04	3.8	140	1.8	0.07	0.6	<0.1	2.6
ZKW 3	1.3	6.4	0.05	3.6	160	2.9	0.06	0.8	<0.1	6.6
ZKW 4	2.3	8.5	0.05	4.8	140	2.1	0.05	0.5	<0.1	4.1
ZKW 5	1.7	10.1	0.04	5.4	80	1.8	0.03	0.4	0.07	bdl
ZKW 6	2.3	8.0	0.04	4.4	420	1.9	0.05	0.5	0.27	bdl

**Table 4 biology-10-01242-t004:** Heavy metals in the Zakawie topsoil fractions.

Sample Number	Fraction (mm)	Zn	Pb	Fe	Mn	Cd	Tl
		(mg·kg^−1^)					
ZK1	>0.71	26,500	16,550	51,300	4100	190	bdl
	>0.355	27,950	19,050	69,150	3800	110	bdl
	>0.180	31,750	20,850	84,150	5100	110	bdl
	>0.09	34,700	30,050	87,200	5550	120	bdl
	>0.045	38,750	40,700	99,250	8750	180	bdl
	<0.045	12,450	12,900	31,700	2550	55	bdl
	>0.71	72,000	21,300	48,050	1350	315	9.9
	>0.355	37,050	10,950	34,100	550	110	8.3
ZK2	>0.180	40,400	15,600	41,350	900	110	bdl
	>0.09	63,900	47,600	72,100	1900	190	bdl
Zk2	>0.045	25,150	10,4400	50,300	1700	190	bdl
	<0.045	36,900	64,750	39,400	1500	180	bdl
	>0.71	13,900	6850	33,450	900	80	bdl
	>0.355	7200	4150	17,600	500	35	bdl
ZK3	>0.180	7800	4650	20,100	600	40	bdl
	>0.09	13,200	8100	34,650	850	70	bdl
Zk3	>0.045	14,050	11,050	41,500	1050	70	bdl
	<0.045	10,300	6700	40,200	900	50	bdl
	>0.355	45,200	2900	61,050	3600	200	bdl
ZK4	>0.180	52,850	2500	52,650	2600	195	bdl
	>0.09	94,650	2900	58,850	2150	345	bdl
Zk4	>0.045	11,1150	2750	62,300	2450	495	bdl
Zk4	<0.045	1400	2250	59,200	2900	380	bdl

**Table 5 biology-10-01242-t005:** Primary and secondary minerals in rhizosphere from historic washing plants in the Zakawie area.

Phases	Minerals	Relative Frequency
Barren phases from Triassic rock	K aluminosilicates, Na aluminosilicates, and others from the group of clay minerals Silica SiO_2_ Ca carbonate-calcite CaCO_3_ K-feldspar and Na-feldspar	+++++ +++ ++ +
Primary phases from Zn-Pb ores	Ca-Mg carbonate-dolomite CaMg(CO_3_)_2_ Ankerite-Ca(Fe,Mg,Mn)(CO_3_)_2_ Barite BaSO_4_	+++ + +
Secondary phases formed in the oxidation process	Pb carbonate-cerussite PbCO_3_ Zn carbonate-smithsonite ZnCO_3_ Monheimite-ZnFeCO_3_ Fe (hydro)oxides-goethite α FeO(OH) and lepidokrokite β FeO(OH) Zn silicate-hemimorfite Zn_4_Si_2_O_7_(OH)_2_ H_2_O Fe-Mn oxides, Mn oxides, e.g., chalcophanite ZnMn_3_O_7_⋅3H_2_O, birnessite group, [Na,Ca,Mn(II)]Mn_7_O_14_⋅2.8H_2_O, and amorphous Mn oxides Ti oxides Ca sulphate-gipsum CaSO_4_ 2H_2_O Pyromorfite (Pb_5_[Cl(PO_4_)_3_])	++ + x +++ + + x x x
Minor phases	Quartz SiO_2_ Magnetite Fe_3_O_4_, Ti oxide-rutile TiO_2_ and ilmenite FeTiO_3_ Ce,La phosphates with RRE	+++ + + +

## Data Availability

On reasonable request, all data can be received from the corresponding authors.

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
