# Peer review of "Vegetation and Environmental Changes on Contaminated Soil Formed on Waste from an Historic Zn-Pb Ore-Washing Plant"

_biology, 2021, doi:10.3390/biology10121242_

Round 1
Reviewer 1 Report
After reviewing the previous version of the manuscript, I find that it is much improved.
However, I feel that the description of vegetation dynamics is still lacking. Although much information concerning what vegetation type converted into what is provided, this is described mostly verbally. It would be much nicer to see this information as a table, with rows indicating 1999 vegetation type and columns 2019 (or vise-versa), with each cells indicating the amount of percentage of are converted. The discussion of these results is very weak, not yielding any clear insight concerning causes, desired trajectories, indicator species or anything of the sort.
Similar problems appear in the rhizosphere results. There is little discussion of differences among succession stages, for example.
This leaves us with a very nicely performed research that does not deliver strong useful conclusions.
Author Response
Authors: First of all, we would like to thank your reviewer very much for his very precise and constructive comments. there is no doubt that they make the paper look better.
Reviewer: After reviewing the previous version of the manuscript, I find that it is much improved. However, I feel that the description of vegetation dynamics is still lacking. Although much information concerning what vegetation type converted into what is provided, this is described mostly verbally. It would be much nicer to see this information as a table, with rows indicating 1999 vegetation type and columns 2019 (or vise-versa), with each cells indicating the amount of percentage of are converted. The discussion of these results is very weak, not yielding any clear insight concerning causes, desired trajectories, indicator species or anything of the sort.
Authors: Thank you for this interesting comment. In the new version of the article, we have included a figure that makes the results of the research easier to read.
Reviewer: Similar problems appear in the rhizosphere results. There is little discussion of differences among succession stages, for example.
Authors: Thank you for this comment. We improved the new version of the manuscript with supplements to the discussion on the issues indicated by the Reviewer. A piece of the text is inserted into the text (tracking in the manuscript).
Reviewer 2 Report
Dear Authors
The current study entitled “Vegetation-and environmental changes on contaminated soil formed on waste from a historic Zn-Pb ore-washing plant” is good. For a better understanding in-depth, it is a need for time to work on this topic. Furthermore, the achievement of potential benefits by using current technology is also dependent on the extensive research work for more exploration. Although previous suggestions are incorporated, yet the article needs more revision before publication. The major concerns are:
1-Avoid using keywords as main title words i.e., soil contamination, zinc and lead.
2- It would be better to give values in percent if values are high. i.e., 85360 mg kg-1 = 8.536%
3- Please provide reference and exact value. General statements (one hectare to more than a dozen) are making the article weak. i.e., "These wastes take up relatively small areas, ranging from one hectare to more than a dozen."
4- Please specify which geochemical transformation in the statement "The pace of development of these ecosystems depends 87 on the degree of geochemical transformation of minerals and potentially toxic metals in 88 the environment." Also, provide suitable reference
Author Response
Reviewer: The current study entitled “Vegetation-and environmental changes on contaminated soil formed on waste from a historic Zn-Pb ore-washing plant” is good. For a better understanding in-depth, it is a need for time to work on this topic. Furthermore, the achievement of potential benefits by using current technology is also dependent on the extensive research work for more exploration.
Authors: Thank you very much for your appreciation of the research problem and its relevance to contemporary environmental research.
Reviewer: Although previous suggestions are incorporated, yet the article needs more revision before publication. The major concerns are:
1-Avoid using keywords as main title words i.e., soil contamination, zinc and lead.
Authors: We thank the Reviewer for all valuable comments on this and the previous version of the manuscript. Thank you. We changed the keywords.
Reviewer:
2- It would be better to give values in percent if values are high. i.e., 85360 mg kg-1 = 8.536%
Authors: We used everywhere mg-kg-1 because it is easy to compare with other elements (with low concentrations). Therefore, it seems to us that this unit should remain.
Reviewer:
3- Please provide reference and exact value. General statements (one hectare to more than a dozen) are making the article weak. i.e., "These wastes take up relatively small areas, ranging from one hectare to more than a dozen."
Authors: Thank you. We made changes to this attribute throughout this article. We cited the paper where can be found more detailed information about the size of heaps and the range of distributions. They are papers with 26 and 29 - citations in this paper.
Reviewer:
4- Please specify which geochemical transformation in the statement "The pace of development of these ecosystems depends on the degree of geochemical transformation of minerals and potentially toxic metals in the environment." Also, provide suitable reference
Authors: Thank you. We have supplemented this point in the text. Provided suitable references: as [2,5-6] and [11] – cited in an introduction.
Round 2
Reviewer 1 Report
The revisions made to the manuscript are satisfactory. Except for some minor English editing, I suggest only two small corrections to the Supplementary Table: 1. Replace the word “genera” (plural) with “genus” (singular). 2. Since none of the species has changed its life form, having two life form columns is redundant. Moreover, you can remove both “observation period” columns and simply leave the two “life form” columns, indicating that if no life form indication is given for a specific year, then the species was not present.
Author Response
We would like to thank the reviewer once again for the right comment.
All comments on the Supplementary Table have been taken into account and are highlighted in red (in the new version of the table).
A colleague, who is a native speaker, has re-read the text and made quite minor changes, claiming that it is OK.
I hope that some style will be checked while waiting for English.
Reviewer 2 Report
Dear Authors
I am satisfied with the changes.
Author Response
We would like to thank the reviewer once again for the right comment during the revision of the first and second versions.